# Prodigiosin from Serratia Marcescens in Cockroach Inhibits the Proliferation of Hepatocellular Carcinoma Cells through Endoplasmic Reticulum Stress-Induced Apoptosis

**DOI:** 10.3390/molecules27217281

**Published:** 2022-10-26

**Authors:** Jie Wang, Hancong Liu, Liuchong Zhu, Jingyi Wang, Xiongming Luo, Wenbin Liu, Yan Ma

**Affiliations:** 1School of Biosciences & Biopharmaceutics, Guangdong Pharmaceutical University, Guangzhou 510000, China; 2Guangdong Provincial Key Laboratory of Pharmaceutical Bioactive Substances, 280 Wai Huan Dong Road, Guangzhou Higher Education Mega Center, Guangzhou 510000, China

**Keywords:** PG, endoplasmic reticulum stress, apoptosis, hepatocellular carcinoma

## Abstract

Hepatocellular carcinoma (HCC) is the most common primary liver malignant tumor, and the targeted therapy for HCC is very limited. Our previous study demonstrated that prodigiosin(PG), a secondary metabolite from Serratia marcescens found in the intestinal flora of cockroaches, inhibits the proliferation of HCC and increases the expression of CHOP, a marker protein for endoplasmic reticulum stress (ERS)-mediated apoptosis, in a dose-dependent manner. However, the mechanisms underlying the activity of PG in vivo and in vitro are unclear. This study explored the molecular mechanisms of PG-induced ERS against liver cancer in vitro and in vivo. The apoptosis of hepatocellular carcinoma cells induced by PG through endoplasmic reticulum stress was observed by flow cytometry, colony formation assay, cell viability assay, immunoblot analysis, and TUNEL assay. The localization of PG in cells was observed using laser confocal fluorescence microscopy. Flow cytometry was used to detect the intracellular Ca^2+^ concentration after PG treatment. We found that PG could promote apoptosis and inhibit the proliferation of HCC. It was localized in the endoplasmic reticulum of HepG2 cells, where it induces the release of Ca^2+^. PG also upregulated the expression of key unfolded response proteins, including PERK, IRE1α, Bip, and CHOP, and related apoptotic proteins, including caspase3, caspase9, and Bax, but down-regulated the expression of anti-apoptotic protein Bcl-2 in liver cancer. Alleviating ERS reversed the above phenomenon. PG had no obvious negative effects on the functioning of the liver, kidney, and other main organs in nude mice, but the growth of liver cancer cells was inhibited by inducing ERS in vivo. The findings of this study showed that PG promotes apoptosis of HCC by inducing ERS.

## 1. Introduction

In recent years, the incidence of primary liver cancer has been increasing worldwide. Hepatocellular carcinoma (HCC) has the highest incidence among primary liver malignancies, and its morbidity and mortality are roughly equivalent [1,2]. The molecular mechanism of HCC formation is very complex, and multiple signaling pathways are involved. These signaling pathways include the PI3K-AKT-mTOR signaling pathway, HGF/C-Met signaling pathway, Wnt/β-catenin pathway, and Notch pathway. These pathways interact with each other and are closely related to the proliferation, survival, and metastasis of liver tumor cells [3,4,5]. Research on the molecular mechanism of liver cancer development can reveal new targets for liver cancer treatment without causing major side effects.

Endoplasmic reticulum stress (ERS) refers to nutrient deficiency, pH change, hypoxia, or oxidative stress, which disrupts the folding function of the endoplasmic reticulum, causing excessive accumulation of poorly folded proteins. The poorly folded protein enters the endoplasmic reticulum, initiates unfolded protein response (UPR), and participates in several tumor biological processes, including metastasis, proliferation, and drug resistance of tumor cells. The proteins can also induce inflammatory immune responses but inhibit antitumor immune responses. ERS is present in many human solid tumors, including liver cancer, lung cancer, nasopharyngeal cancer, breast cancer, etc., and is closely related to drug resistance and poor prognosis of cancers [6,7]. ERS further promotes the release of inflammatory factors from tumor-associated macrophages and the infiltration of inflammatory cells. This forms a vicious cycle of ERS, inflammation, and tumorigenesis. However, very high or prolonged endoplasmic reticulum stress promotes the apoptosis of tumor cells by regulating the cellular survival pathway that inhibits cell survival and directly removes cells that have been irreversibly damaged [8,9].

Microbial secondary metabolites are natural products with numerous properties, including antibacterial and antitumor properties. Our previous study revealed that Serratia marcescens in the intestinal flora of cockroaches procure Prodigiosin (PG), a metabolite with a strong antitumor effect against nasopharyngeal carcinoma and cervical cancer cells [10]. PG is a natural red-pigmented secondary metabolite, mainly produced by Serratia marcescens and actinomycetes. The structure of the bioactive agent differs among different bacteria, but all have a 4-methoxy-2-2 pyrrole ring structure. This metabolite possesses antibacterial, anti-fungal, anti-malaria immunosuppressive, and antitumor properties. PG abnormally activates the Wnt/β-catenin signaling pathway, reverses the abnormal expression of survivin in liver cancer cells, and promotes the apoptosis of these cells [11]. PG also inhibits the growth of human oral squamous cell carcinoma cells by targeting the autophagic cell death pathway. Hosseini’s research results showed that PG had a great affinity for the anti-apoptotic member MCL-1 of the BCL-2 family, and it could activate the mitochondrial apoptosis pathway by destroying the MCL-1/BAK complex to cause apoptosis in melanoma cells [12]. Therefore, it is necessary to clarify the role of different molecular phenotypes in the PG-induced apoptosis of malignant tumor cells and the molecular mechanisms underlying this process, which can reveal new and more effective targets for treating tumors. Our previous research revealed that PG increased the expression of CHOP, a marker protein for ERS-mediated apoptosis of the liver cancer cells, in a concentration-dependent manner. ERS can have the dual effects of either promoting or inhibiting apoptosis, implying that the antitumor activity of PG is related to ERS. In the present study, we performed in vitro and in vivo experiments to further investigate the effects of PG on genetically diverse HCC using HepG2 and BEL7402 cells. The molecular mechanism underlying the effect of PG on HCC was also investigated.

## 2. Results

### 2.1. PG Inhibits the Proliferation and Viability but Promotes the Apoptosis of Hepatocellular Carcinoma Cells

The Cell Counting Kit-8 (CCK-8) assay revealed that (Figure 1A,B) PG inhibits the proliferation of HepG2 and BEL-7402 cells in a time and dose-dependent manner. The CCK-8 assay demonstrated that the IC50 values of PG on HepG2 and BEL7402 cells at 48 h were 1.01 μg/mL and 2.69 μg/mL, respectively. The inhibitory effect of PG on the viability of HepG2 was stronger than that of Bel7402. Colony formation assay further showed that PG had a significant effect on the colony formation property of HepG2 and Bel7402 (Figure 1C and Figure 2D). Flow cytometry showed that PG arrested the cell cycle of HCC in the G0/G1 phase (Figure 2A) and the toxicity of PG to HCC increased with the increase in concentration (Figure 2B).

### 2.2. The Effect of Inhibiting ERS on Apoptosis of HCC Induced by PG

The effect of PG on apoptosis of HepG2 and Bel7402 was detected by TUNEL assay. The results showed that the fluorescence intensity of cells increased with the concentration of PG. However, ERS-inhibitor 4-Phenylbutyric acid (4-PBA) pretreatment reduced the fluorescence intensity of cells (Figure 3A,B). Western blotting revealed that PG treatment significantly increased the expression of Bax, caspase3, and caspase9 but decreased that of Bcl-2. However, 4-PBA pretreatment reversed the abnormal expression of the above proteins (Figure 3C,D). The results showed that PG regulated the expression of apoptotic and anti-apoptotic proteins in a dose-dependent manner, and inhibition of ER stress partially reversed the PG-induced apoptosis of HepG2 and Bel7402.

### 2.3. Effects of PG on ERS in HepG2 and Bel7402 Cells

Calnexin is an endoplasmic reticulum-bound chaperone protein and an endoplasmic reticulum stress-related unfolded response protein. It is a molecular chaperone that mainly participates in the folding and processing of protein nascent peptide chains. The cellular localization of PG in HepG2 cells was observed using a confocal laser fluorescence microscope (Figure 4A). We found that the correlation between PG and calnexin was stronger with the increase in PG concentration, indicating that more PG was localized in the ER. The endoplasmic reticulum also stores Ca^2+^. Disrupting the Ca^2+^ balance in the endoplasmic reticulum induces ERS. The fluorescence intensity of HepG2 and Bel-7402 treated with different concentrations of PG was detected by flow cytometry (Figure 4B). We found that PG increased the expression of signal strength in a dose-dependent manner, suggesting that PG increases the Ca^2+^ in HepG2 and Bel-7402 cells. Western blot further revealed that PG increased the expression of PERK, IRE1α, CHOP, and endoplasmic reticulum chaperone protein Bip in HepG2 and Bel7402 cells via the unfolded protein response signaling pathway (Figure 5A,B). PG treatment increased the protein expressions of PERK, IRE1α, CHOP, and Bip, but 4-PBA pretreatment for 2 h reversed this phenomenon. The expression of ATF4, CHOP, XBP1, and Bip genes in HepG2 and Bel7402 after PG treatment was detected by real-time fluorescence quantitative PCR (Figure 5C). We found that PG increased the transcription of mRNAs. However, 4-PBA pretreatment reversed this phenomenon.

### 2.4. PG Reduced the Proliferation of HCC In Vivo

The nude mice model of HCC HepG2 was successfully constructed, and the subcutaneously transplanted tumor was dissected and weighed (Figure 6A). The tumor volume and weight were significantly smaller in the treatment group than that in the normal saline control group. The inhibitory rates of PG (2.5 mg/kg), PG (5 mg/kg), and the 4-PBA in combination with PG (5 mg/kg) were 35.6% ± 0.493、47.0% ± 0.419 and 45.9% ± 0.169, respectively. The biosafety of PG was also evaluated in vivo. PG had no significant effect on liver and kidney function (Figure 6B). The effect of PG on the functioning of the heart, liver, spleen, lung, and kidney was analyzed based on histomorphological analyses (Figure 7). Hematoxylin–eosin staining (HE staining) results showed well-arranged and dense myocardial myofilaments in the liver and kidney of nude mice were dense; the hepatic lobules and spleen nodules were normal with clear outlines; the alveoli and glomeruli were also generally normal. These results indicated that PG had no obvious toxic effect on liver and kidney function and other main organs of nude mice.

### 2.5. The Apoptosis-Inducing Effect of PG on Transplanted Tumors in Nude Mice

The effect of PG on the apoptosis of tumor tissue HepG2 cells was analyzed using the TUNEL assay (Figure 8A). The green fluorescence signal around the nuclei of tumor tissue increased with PG concentration. To observe the effect of PG-induced ERS on the apoptosis of tumor tissue, the expression of Bip, PERK, and caspase3 was detected using immunohistochemical analysis (Figure 8B). Compared with the control group, we found that PG increased the expression of Bip, PERK, and caspase3 in a concentration-dependent manner. However, 4-PBA pretreatment decreased the expression of these proteins. The expression of key proteins in the ERS pathway and apoptosis-related proteins in tumor tissues after PG treatment displayed a similar trend (Figure 8C).

## 3. Discussion

PG can promote the apoptosis of various tumor cells, such as lung, colon, oral, nasopharyngeal, breast, and liver cancer cells [13]. In recent years, the use of PG in combination with other cancer therapies has also been explored as a promising strategy for cancer treatment. It has been reported that PG inhibits autophagy to sensitize colorectal cancer cells to 5-fluorouracil, and the combination therapy significantly reduces the viability of cancer cells, partly through caspase-dependent apoptosis [14]. Furthermore, Shian-ren Lin [15] found that PG can induce autophagy, promote adriamycin influx and sensitize oral squamous carcinoma cells. A derivative of PG named obatoclax could promote the release of cytochrome C from the mitochondria of isolated leukemia cells. The combination of obatoclax and Ara C could synergistically induce apoptosis in leukemia cell lines and primary Acute Myelocytic Leukemia (AML) samples [16]. Several phase I and II clinical studies used a derivative of PG named obatoclax for cancer therapy on different patients [17,18,19,20]. Jelena’s study showed that PG and its Br derivatives showed anticancer potential against all tumor cell lines and induced apoptosis, but their selectivity to healthy cell lines was low. The greater lipophilicity of Br derivatives of PG made them good targets for further structural optimization [12].

Additional evidence suggests that ERS-induced cellular dysfunction and cell death are primary contributors to many diseases [21], making modulation of the ERS pathway a potentially attractive therapeutic target. Moreover, this study further explored the mechanism of PG-mediated ERS-induced apoptosis of liver cancer cells and provided a theoretical basis for improving the anticancer effect of the drug.

In this study, the CCK-8 assay demonstrated that the IC50 values of PG on HepG2 and BEL7402 cells at 48 h were different. The optimal drug concentration was selected according to the IC50 value in subsequent experiments. The CCK-8 and cell colony formation assay results showed that PG could significantly inhibit the proliferation of HepG2 and Bel7402 cells, and HepG2 was more sensitive to PG than Bel7402 cells, which may be due to their different genetic backgrounds. Flow cytometry showed that PG blocked the cell cycle of HCC in the G0/G1 phase and induced apoptosis of HepG2 and Bel7402 cells in a dose-dependent manner. Another study showed that PG could reduce the expression of the anti-apoptotic protein survivin in liver cancer cells and activate caspase3, resulting in cell death [11]. However, its specific anti-hepatocellular carcinoma target and signal pathway are not clear.

In this study, we further observed the effect of PG on hepatocellular carcinoma cell apoptosis through ERS using the ERS inhibitor 4-PBA. TUNEL staining showed that the apoptosis of HCC cells decreased after 4-PBA pretreatment, and Western blot showed that the 4-PBA pretreatment reversed the expression change of the apoptotic proteins Bax, caspase3, caspase9, and anti-apoptotic protein Bcl-2. It has been reported that PG-induced apoptosis of tumor cells mainly occurs via the regulation of the mitochondrial apoptosis pathway. Cytochrome C released in the mitochondrial apoptosis pathway can activate caspase9 to shear apoptosis pathway caspase3, thus inducing apoptosis [22]. In melanoma cells, PG could bind to the BH3 domain and activate the mitochondrial apoptosis pathway by disrupting the McL-1/BAK complex, an anti-apoptotic member of the Bcl-2 family [23]. Our results suggest that PG induced the apoptosis of hepatocellular carcinoma cells by triggering ERS. Confocal laser fluorescence microscopy revealed that the action location of PG was co-located with calnexin in the endoplasmic reticulum of HepG2 cells. Further, flow cytometry detected an increase in Ca^2+^ concentration in hepatocellular carcinoma cells treated with PG. It has been reported [24] that acute release of Ca^2+^ in the endoplasmic reticulum can inhibit protein folding and trigger multiple signaling mechanisms to mediate mitochondrial apoptosis pathways. This study revealed that the endoplasmic reticulum was the active site of PG action in hepatocellular carcinoma cells, indicating that the mechanism of this substance is closely related to ER. Western blot analysis was then used to detect the expression of axon-related proteins in the unfolded protein signal branches PERK-elF2α-ATF4-CHOP and IRE1α- XBP11-CHOP. The results showed that PG upregulated the expression of PERK, IRE1α, Bip, and CHOP, but these expressions decreased significantly after 4-PBA pretreatment. The expression levels of the downstream transcription factors ATF4, CHOP, and XBP1 genes and the ER chaperone Bip gene of unfolded protein signal branch were also upregulated following PG exposure but downregulated by 4-PBA pretreatment. ERS-induced apoptosis occurs via the mitochondrial apoptotic pathway, which is regulated by the Bcl-2 protein family [25]. CHOP is one of the key pro-apoptotic factors of UPR, and its transcription is positively regulated by the PERK-ATF4 axis. Notably, CHOP can induce apoptosis by promoting the transcription of Bax and down-regulating the expression of Bcl-2 [26].

At present, most previous research on the antitumor activity of PG has been performed in vitro, with very few conducted in vivo. Our in vivo study demonstrated that PG has no obvious toxic effects on the main organs and related indicators of liver and kidney function in nude mice but inhibits tumor growth. These findings concur with Obayemi [27], who reported that PG does not cause toxic effects in mice. In our study, the expression of PERK and Bip proteins in the ERS pathway and the related apoptotic protein caspase3 increased with the drug concentration, but the expression levels of these proteins in the combined 4-PBA group decreased compared with the PG alone group. This finding of the in vivo experiments was consistent with the in vitro experiments, but the difference was not significant. Possibly, the difference in the effect of the drug between the in vivo and in vitro experiments was caused by variations in the absorption efficiency of the drug in animals. This study suggests that PG may exhibit anti-HCC activity by acting via ERS.

## 4. Materials and Methods

### 4.1. Cell lines and Laboratory Animals

HepG2 (HCC, ATCC HB-8065) and Bel7402 (HCC, SNL-148) were purchased from Guangdong Provincial Key Laboratory of Pharmaceutical Bioactive Substances. The cells were cultured at 37 °C in RPMI-1640 medium (Gibco) supplemented with 10% fetal bovine serum (Gemini) and a mixture of 1% penicillin and streptomycin (Hyclone) under 5% CO_2_. A total of 36 4–6 weeks old SPF male BALB/C (nu/nu) nude mice weighing 16–20 g were purchased from Guangdong Medical Experimental Animal Center and were reared in an IVC system of the Animal Center of Guangdong Pharmaceutical University. The protocol for animal experiments was approved by the Guangdong Medical Experimental Animal Center (license number SCXK (Guangdong) 2018–0002). The animals were reared at 23–28 °C, at a relative humidity of 30–70%, and a light cycle of 12 h/d.

### 4.2. Antibodies and Reagents

Antibodies used in this study were anti PERK (Cell Signaling, D11A8), IRE1α (Cell Signaling, 14C10), Bcl-2 (Cell Signaling, D55G8), Bip (Cell Signaling, C50B12), CHOP (Cell Signaling, L63F7), Bax (Cell Signaling, D2E11), caspase3 (Bioworld, BS6428), caspase9 (Bioworld, BS1731), β-Actin (Signaling, 8H10D10), Anti-GAPDH (Genomics, 5A12), and anti-Alexa Fluor 488 Labeled Goat Anti-Rabbit IgG (H+L) (Biyuntian, A0423). All these antibodies were purchased from Shanghai Biyuntian Biological Company. PG was isolated from Serratia marcescens, strain WA12-1-18, in the intestinal tract of cockroaches. The metabolite was extracted using 3.0 g/L of PBS from previously described [10]. Flou3-AM fluorescent probe, PVDF membrane, (2-(4-Amidinophenyl)-6-indolecarbamidine dihydrochloride) DAPI, TUNEL apoptosis kit, and ECL kits were purchased from Shanghai Biyuntian Biological Company (Shanghai, China), whereas 4-PBA was purchased from Sigma Aldrich Company (St. Louis, MI, USA). A fluorescent quantitative PCR kit was purchased from Thermo Fisher Scientific (Waltham, MA, USA). Cell cycle and apoptosis kits were purchased from Beijing Sizhengbai Biotechnology Company (Beijing, China).

### 4.3. Cell Viability Assay

HepG2 and Bel7402 cells (5×10^3^ cells/mL) in 100 μL cell suspension were seeded into 96-well plates and incubated at 37 °C for 12 h under 5% CO_2_. The cells were treated with different PG concentrations (0.2, 0.4, 0.8, 1.6, 3.2, 6.4, 12.8 and 25.6 μg/mL) for 24, 48, and 72 h until the cell confluence reached about 80%. Each well was then washed twice with PBS. The absorbance at 450 nm was measured by Fully automatic enzyme labeling instrument (BioTek Instruments, Inc. USA) after adding CCK-8 (Biosharp) solution in the wells for a 1.5 h culture.

### 4.4. Colony Formation Assay

HepG2 and Bel7402 cells were seeded into 6-well plates (1000 cells/mL) and cultured at 37 °C for 24 h under 5% CO_2_. The cells were treated with different PG concentrations (0.2, 0.4, 0.8, 1.6, and 3.2 μg/mL) for 1 h. The control group was treated with anhydrous ethanol. The culture was continued for 10 days after replacing the wells with a complete medium. Culturing was stopped when the confluence of the clone community cells in the control group grew above 80%, as observed under a microscope. After discarding the supernatant, the cells were fixed for 20 min using 1 mL of 4% paraformaldehyde. The cells were then stained using 1 mL of 0.1% crystal violet staining solution for 15 min at room temperature. The cells were photographed by digital camera (Nikon D3200, Tokyo, Japan), and the number of clonal colonies was counted. Number of clusters (Fold change to control) was calculated as: (Colony counts experiment group/Colony counts medium control group) × 100%.

### 4.5. Cell Cycle and Apoptosis Assays Using Flow Cytometry

HepG2 and Bel7402 cells (100 μL; 5 × 10^3^ cells/mL) were added to 6-well plates and incubated at 37 °C for 12 h under 5%CO_2_. When the cell confluence was about 80%, the cells were treated for 48 h with different PG concentrations (0.4, 0.8, 1.6, and 3.2 μg/mL). The control group was treated with anhydrous ethanol. The cells were then fixed with precooled 70% ethanol at 4 °C overnight. The cells were stained using 400 μL of propidium iodide staining solution for 30 min at 37 °C before immediately detecting the cell cycle. For apoptosis rate detection, the cells were digested using trypsin without EDTA and suspended in Binding Buffer at a concentration of 1~5 × 10^6^/mL. Thereafter, 100 μL of cell suspension was pipetted into a 5 mL flow tube. Annexin V-EGFP 5 μL was added to the flow tube and incubated in darkness for 5 min. The cells were then stained using 10 μL PI staining solution before adding 400 μL PBS. All sample flow detection was then performed immediately by flow cytometry (FACS Calibur; Becton, Dickinson and Company, Franklin Lakes, NJ, USA).

### 4.6. TUNEL Assay

When the cell confluence reached about 80%, the cells were pretreated with ERS-inhibitor 4-PBA (2 mm) for 2 h. The cells were then treated with different PG concentrations (0.4, 0.8, 1.6, and 3.2 μg/mL) for 48 h. The control group was treated with anhydrous ethanol. HepG2 and Bel7402 cells (3 × 10^4^/mL) were fixed with 4% paraformaldehyde, embedded in liquid paraffin, and stored in an oven at 62 °C for 1.5 h before gradient dewaxing. Thereafter, 50 μL of 20 μg/mL of DNase-free protease K was added to the tissue for 20 min at 30 °C. TUNEL assay was performed following the manufacturer’s instructions. The cell nuclei were then stained using DAPI and observed under a fluorescence microscope (OX31; Olympus, Tokyo, Japan) [28].

### 4.7. Immunoblot Analysis

For immunoblot assay, 20–30 µg of total cell extracts were separated in 10% or 12% SDS-PAGE and transferred onto the PVDF membranes. The membrane was then stained with primary antibodies against Bax, caspase3, caspase9, Bcl-2, PERK, IRE1α, CHOP, and Bip at a 1:1000 dilution. After rinsing using PBS, the membrane we incubated with HRP-conjugated secondary antibodies goat anti-rabbit or anti-mouse was at a 1:2000 dilution. The proteins were detected using enhanced chemiluminescence. 

### 4.8. The Localization of PG in Cells Using a Laser Confocal Fluorescence Microscopy

HepG2 cells (1.5×10^5^/mL) were seeded into laser confocal dishes and cultured with different PG concentrations (0.4, 0.8, and 1.6 μg/mL). After fixing with 4% paraformaldehyde for 20 min, the cells were incubated with 1% BSA for 30 min. The cells were then incubated overnight at 4 °C in calnexin diluted with 1% BSA and followed by Alexa Fluor 488-conjugated goat anti-rabbit secondary antibodies for 1 h. After DAPI staining for 5 min, the cells were washed three times with 1 × PBS, and added the fluorescence quenching agent. The intracellular calnexin protein was stained with green fluorescence. Since the PG itself had a spontaneous red fluorescence, the coincidence of the two can be used to determine co-localization. The cells were then observed and photographed using a laser confocal fluorescence microscope (FV3000; Olympus, Japan). Pearson’s R value was calculated by using image J software (v1.8.0)

### 4.9. The measurement of Ca^2+^ Concentration in Cells Using Flow Cytometry

Briefly, 1 μM fluo-3 AM was added into HCC treated with different PG concentrations (0.4, 0.8, 1.6, and 3.2 μg/mL) for 50 min in the dark. The cells were then suspended in 300 μL PBS and then incubated for 15 min in the dark. The fluorescence intensity was detected using flow cytometry [29].

### 4.10. Real-Time PCR Analysis

HepG2 and Bel7402 cells (3 × 10^5^/mL) were pretreated with 4-PBA for 2 h, and thereafter with different PG concentrations (0.2, 0.4, 0.8, 1.6, 3.2 μg/mL). Anhydrous ethanol was used as the negative control. RNA was extracted from the cells and reverse-transcribed into cDNA. The primer sequences were as follows:

CHOP-forward primer: 5′-GGAGCATCAGTCCCCCACTT-3′,

CHOP-reverse primer: 5′-TGTGGGATTGAGGGTCACATC-3′;

ATF4-forward primer: 5′-GCTAAGGCGGGCTCCTCCGA-3′,

ATF4-reverse primer: 5′-ACCCAACAGGGCATCCAAGTCG-3′;

XBP1-forward primer: 5′-CCTGGTTGCTGAAGAGGAGG-3′,

XBP1-reverse primer: 5′-CCATGGGGAGATGTTCTGGAG-3′;

Bip-forward primer: 5′-TGACATTGAAGACTTCAAAGCT-3′,

Bip-reverse primer: 5′-CTGCTGTATCCTCTTCACCAGT-3′;

GAPDH-forward primer: 5′-GGAGCGAGATCCCTCCAAAAT-3′,

GAPDH-reverse primer: 5′-GGCTGTTGTCATACTTCTCATGG-3′.

### 4.11. In Vivo Tumor Model

HepG2 cells (200 μL; 2 × 10^7^/mL) were inoculated in the right lower axilla of 36 nude mice. The mice were then randomly divided into 6 groups: normal saline group (1‰ ethanol), PG2.5 (2.5 mg/kg), PG5.0 (5 mg/kg), PG5.0 (5 mg/kg) +4-PBA (150 mg/kg), 4-PBA (150 mg/kg) and cis-platinum (DDP) (2 mg/kg). DDP was a positive group. When the tumor volume reached 100 mm^3^, the mice were injected intraperitoneally with saline (1‰ ethanol), PG2.5 (2.5 mg/kg), PG5.0 (5 mg/kg), PG5.0 (5 mg/kg) +4-PBA (150 mg/kg), 4-PBA (150 mg/kg) and DDP (2 mg/kg), once every four days for 24 days. The tumor volume was calculated as follows; TV (mm^3^) = (LW^2^)/2, where TV is the tumor volume, L is the tumor length, and W is the shortest tumor diameter, measured on a vernier scale.

### 4.12. Assessment of Liver and Kidney Function

Blood was collected from the eyeballs of nude mice, and the serum was taken after centrifugation and sent to Jiangsu Enzymatic immunity Industry Co., Ltd. (Yancheng, China) for testing. AST (blood volume: 15 μL), ALT (blood volume: 15 μL), ALP (blood volume: 15 μL), ALB (blood volume: 5 μL), TP (blood volume: 6 μL), CR (blood volume: 5 μL), UA (blood volume: 5 μL), and urea (blood volume: 3 μL) content in the serum was then measured by automatic biochemical analyzer (BK-280; BIOBASE, Jinan, China).

### 4.13. Hematoxylin–Eosin Staining

Tissue sections embedded in paraffin were heated in an oven at 62 °C for 1.5 h, dewaxed in water, and stained with hematoxylin for 5 min and eosin for 5 s. After dehydrating the tissue sections in serial alcohol concentrations, the tissue slices were sealed with neutral gum and photographed under a microscope (SQS-40P, Discover echo, San Diego CA, USA).

### 4.14. Statistical Analysis

Continuous normally distributed data were expressed as means of at least three independent experiments ± SEM/SD. The data were analyzed using SPSS software, V 22.0. Differences between two groups were analyzed using Student’s *t*-test and One-way ANOVA for comparison between multiple groups. GraphPad Prism 8 (GraphPad Software Inc.) was used for the creation. *p* < 0.05 was considered statistically significant.

## 5. Conclusions

In conclusion, this study explored the mechanism of anti-liver cancer action of PG in vitro and in vivo. The findings enrich the current literature on the antitumor mechanism of PG and provide laboratory evidence for its application in precision medicine and personalized therapy in HCC.

## Figures and Tables

**Figure 1 molecules-27-07281-f001:**
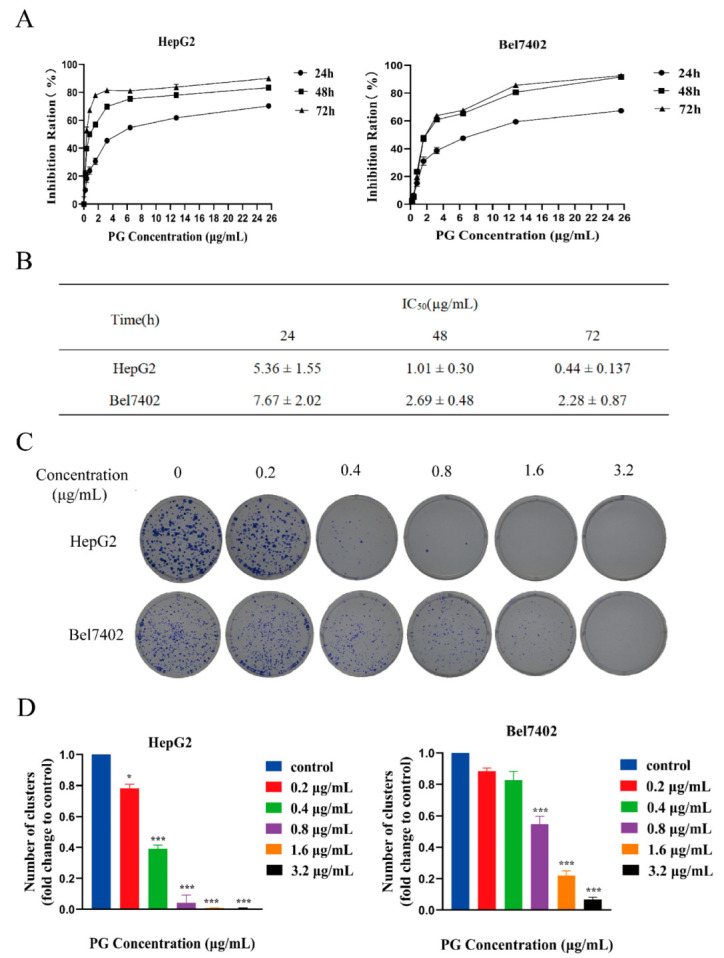
The effects of PG on the proliferation, cell cycle, and apoptosis of HepG2 and BEL7402 cells. (**A**). Inhibition rate of HepG2 and BEL7402 treated by PG for 24 h, 48 h and 72 h, respectively. (**B**). The IC50 values of the cells treated with PG were at 24, 48 and 72 h, respectively. (**C**). The effect of PG on the colony formation property of HepG2 and BEL7402. (**D**) HepG2: PG (1.6 μg/mL), PG (0.8 μg/mL) and PG (0.4 μg/mL) vs. PG (0 μg/mL), * *p* < 0.05; *** *p* < 0.001, *n* = 3; Bel7402: PG (3.2 μg/mL), PG (1.6 μg/mL) and PG (0.8 μg/mL) vs. PG (0 μg/mL), *** *p* < 0.001, *n* = 3.

**Figure 2 molecules-27-07281-f002:**
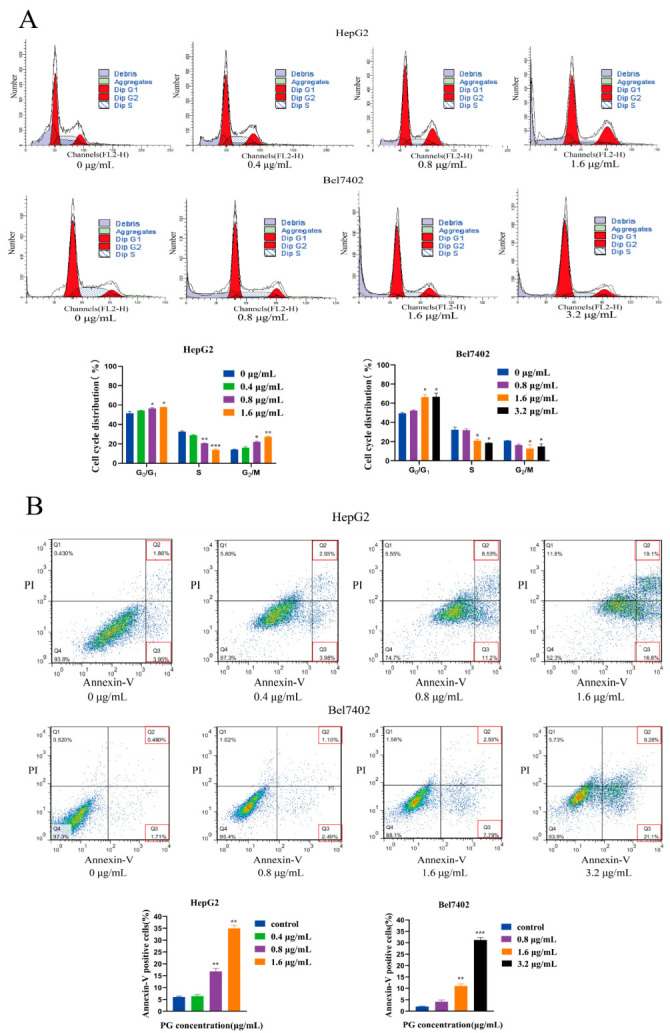
(**A**). Effect of PG on HepG2 and BEL7402 cell cycle. (**B**). Effect of PG on apoptosis of HePG2 and BEL7402 cells. The combined percentage of cells in Q2 and Q3 represents Annexin V-positive cells. HepG2: PG (1.6 μg/mL), PG (0.8 μg/mL) and PG (0.4 μg/mL) vs. PG (0 μg/mL), * *p* < 0.05, ** *p* < 0.01, *n* = 3; Bel7402: PG (3.2 μg/mL), PG (1.6 μg/mL) and PG (0.8 μg/mL) vs. PG (0 μg/mL), * *p* < 0.05, ** *p* < 0.01 and *** *p* < 0.001, *n* = 3.

**Figure 3 molecules-27-07281-f003:**
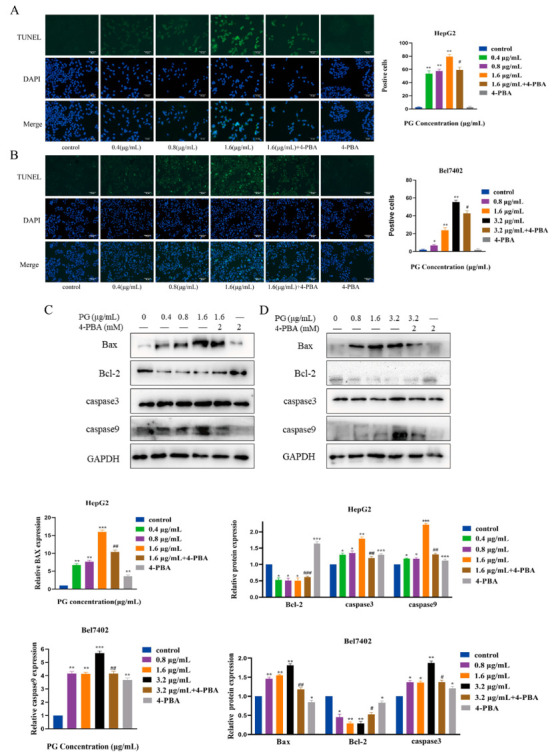
Induction of apoptosis by PG treatment of HCC cells. (**A**). TUNEL assay for the effect of 4-PBA pretreatment on the apoptotic effect of PG on HepG2 cells. (**B**). TUNEL assay for the effect of 4-PBA pretreatment on the apoptotic effect of PG on Bel7402; (**C**). Western blot for the effect of 4-PBA pretreatment on the expression of apoptotic proteins in HepG2; (**D**). Western blot for the effect of 4-PBA pretreatment on the expression of apoptotic proteins in Bel7402 cells. The treatments included PG (0.4 μg/mL, 0.8 μg/mL, 1.6 μg/mL, 3.2 μg/mL) vs. 0 μg/mL of PG; 1.6 μg/mL of PG vs. 1.6 μg/mL of PG + 4-PBA; 3.2 μg/mL PG vs. 3.2 μg/mL of PG + 4-PBA; * *p* < 0.05, ** *p* < 0.01, *** *p* < 0.001, # *p* < 0.05, ## *p* < 0.01, ### *p* < 0.001, *n* = 3.

**Figure 4 molecules-27-07281-f004:**
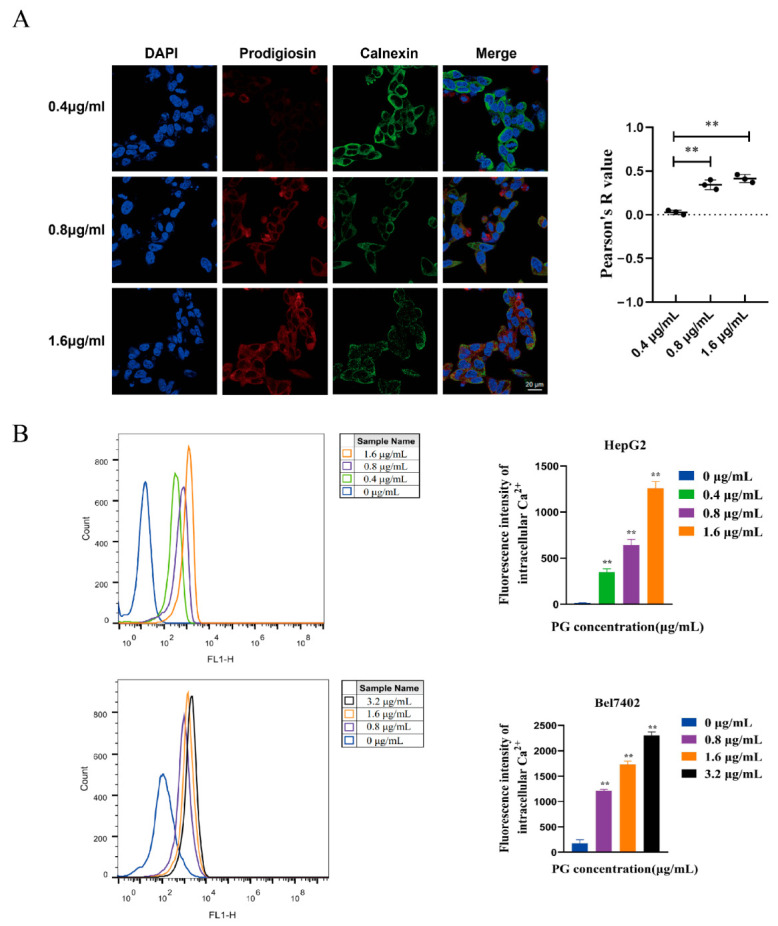
The effects of PG on ERS pathway in HepG2 and Bel7402 cells. (**A**). Localization of PG in HepG2 cells as observed under confocal laser fluorescence microscopy; (**B**). Flow cytometry for the effect of PG on Ca^2+^ concentration in HepG2 and Bel7402 cells. The treatments included PG (0.4 μg/mL, 0.8 μg/mL, 1.6 μg/mL, 3.2 μg/mL) vs. 0 μg/mL of PG, ** *p* < 0.01, *n* = 3.

**Figure 5 molecules-27-07281-f005:**
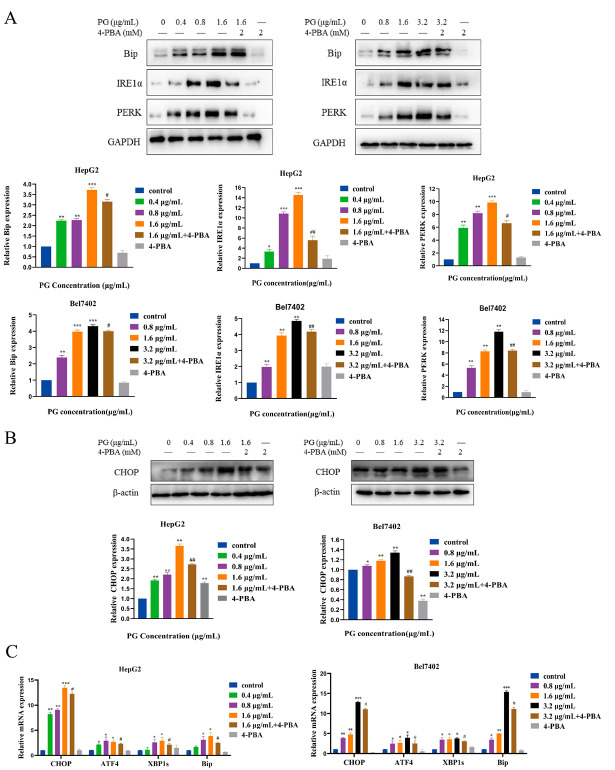
(**A**). Western blot assay on the expression of Bip, IRE1α, and PERK in HepG2 and Bel7402 cells; (**B**). Western blot assay for the expression of CHOP in HepG2 and Bel7402 cells; (**C**). RT-qPCR effect of 4-PBA pretreatment on the effect of PG on the expression of Bip, CHOP, ATF4, and XBP1 mRNA in HepG2 and Bel7402 cells. The treatments included PG (0.4 μg/mL, 0.8 μg/mL, 1.6 μg/mL, 3.2 μg/mL) vs. 0 μg/mL of PG; 1.6 μg/mL of PG vs. 1.6 μg/mL of PG + 4-PBA; 3.2 μg/mL PG vs. 3.2 μg/mL of PG + 4-PBA; * *p* < 0.05, ** *p* < 0.01, *** *p* < 0.001, # *p* < 0.05, ## *p* < 0.01, n = 3.

**Figure 6 molecules-27-07281-f006:**
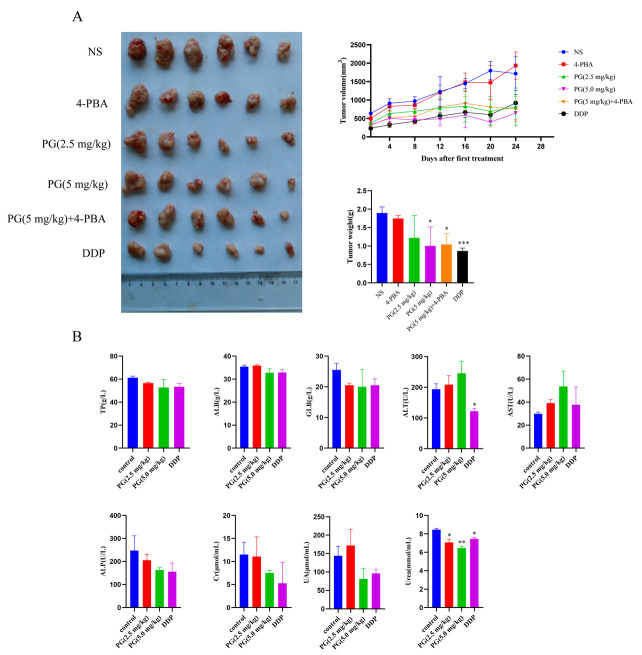
The effect of PG on the proliferation of hepatocellular carcinoma in vivo. (**A**). Effects of PG on the volume and weight of subcutaneously transplanted tumors; (**B**). The effect of PG on the liver and kidney function of nude mice with HepG2. The levels of total aspartate aminotransferase (AST), alanine aminotransferase (ALT), alkaline phosphatase (ALP), albumin (ALB), total protein (TP), creatinine (CR), uric acid (UA) and urea in the blood represent the liver and kidney function of nude mice. PG (2.5 mg/kg), PG (5 mg/kg), PG (5 mg/kg) +4-PBA, DDP vs. NS, * *p* < 0.05, ** *p* < 0.01, *** *p* < 0.001, *n* = 6.

**Figure 7 molecules-27-07281-f007:**
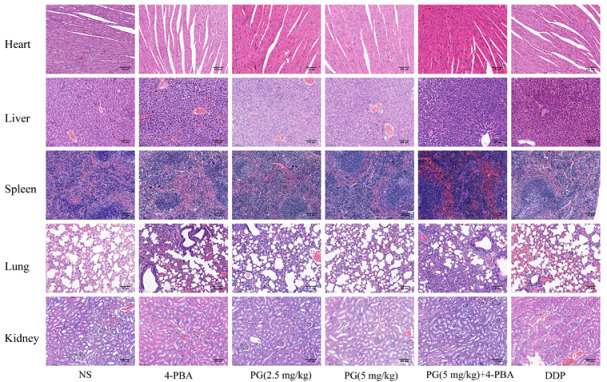
HE staining of heart, liver, spleen, lung, and kidney tissues of nude mice.

**Figure 8 molecules-27-07281-f008:**
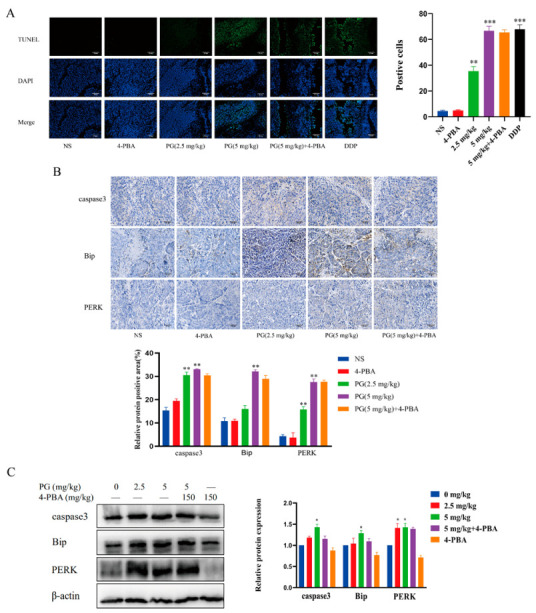
The effect of PG on the apoptosis in the tumor xenografts of HepG2 HCC model. (**A**). TUNEL assay for the apoptotic property of PG on HCC; (**B**). Immunohistochemical analysis for the effect of PG on the expression of caspase3, Bip and PERK in HCC tissue; (**C**). Western blot for the effect of PG on the expression of Bip, PERK, and caspase3 proteins in HCC tissues. PG (2.5 mg/kg), PG (5 mg/kg) vs. PG (0 mg/kg), * *p* < 0.05, ** *p* < 0.01,****p* < 0.001, *n* = 3; PG (5 mg/kg) vs. PG (5 mg/kg) + 4-PBA, *n* = 3.

## Data Availability

Not applicable.

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
