# Peer review of "Prodigiosin from Serratia Marcescens in Cockroach Inhibits the Proliferation of Hepatocellular Carcinoma Cells through Endoplasmic Reticulum Stress-Induced Apoptosis"

_molecules, 2022, doi:10.3390/molecules27217281_

Round 1

Reviewer 1 Report

This paper addresses the effects of prodigiosin (PG) on liver tumor growth. Although I have an interest in anti-cancer molecules, the methods used in the paper lie completely outside my sphere of expertise. Overall the paper is well-written and the standard of English very high, but as a layman I am completely baffled by terms like CCK-8, 4-PBA, DAPI, DPP and many others, which are nowhere explained. Above all else, a scientific paper should be understandable.

Line 19 in the Abstract needs editing.

One line of explanation of the HepG2 and BEL-7402 cell lines would be helpful, giving the tissue type and explaining the key differences. On line 212 the paper mentions melanoma cells, which appear nowhere else in the paper. The TUNEL assay is unknown to me.

What is 4-PBA, and why was it added? What effect was predicted, and did this happen?

The statistics demonstrating significance are sometimes missing. For example it is stated on Page 6, line 144, that the tumor volume and weight were significantly smaller in the treatment group, but how many tumors were examined? What are the standard deviations on the inhibition rates?

Figure 4B is completely unexplained, with the legend simply repeating a line of the text. What does this figure show, and why is it important? I do not understand a single acronym on the vertical axes of these graphs.

The importance of the paper seems to be the fact that it demonstrates anti-tumor activity in vivo, while leaving healthy tissue unharmed. The major conclusion is that PG exerts its effect through stress of the endoplasmic reticulum. PG increases ER stress, and apparently therefore upregulates the expression of Bip, an ER chaperone. 4-PBA seems to reduce ER stress, and also the expression of ER chaperones. It is not described what efforts have been made to determine the cellular target(s) of PG, and whether it works in direct opposition to 4-PBA.

Reading the very extensive literature on PG, I came across a paper (https://doi.org/10.1158/0008-5472.CAN-07-1919) by Konopleva and colleagues from 2008 which answered several of the questions I found myself asking, which provides a simple mechanism for the cellular effects of PG derivatives. I think the authors of the present manuscript could have saved me some time by citing it.

On line 211, the authors state "Cytochrome C released in the mitochondrial apoptosis pathway can activate caspase9 to shear apoptosis pathway caspase3, thus inducing apoptosis [20]." Looking at the paper cited (Hosseini et al, 2013), we find in the Abstract nothing about cytochrome c or caspases. Quoting Hosseini, we find "our results indicate a large affinity of prodigiosin for MCL-1, an anti-apoptotic member of the BCL-2 family." Wonderful! Lights switch on in my head! We have a small molecule binding to a known protein of known function, and my feet have found solid ground. Citing this paper early in the Introduction, at the end of a sentence which accurately reflects its contents, would make this paper a far, far easier read.

Clearly the PG family of molecules has been undergoing clinical trials for many years, and some mention could be made of the current state of affairs. Why have trials stopped, and what can the authors do to bring new derivatives to the clinic? Above all, why does PG exert an effect on cancer cells, and not healthy ones?

Overall the papers cited are a very small selection from a large field, and one wonders why recent relevant papers from the same journal (such as doi: 10.3390/molecules27123729) are not mentioned. Expanding the introduction would give the authors ample scope to describe the recent progress in this area. For the record, I have never published anything related to PG.

Author Response

Point 1: This paper addresses the effects of prodigiosin (PG) on liver tumor growth. Although I have an interest in anti-cancer molecules, the methods used in the paper lie completely outside my sphere of expertise. Overall the paper is well-written and the standard of English very high, but as a layman,I am completely baffled by terms like CCK-8, 4-PBA, DAPI, DDP and many others, which are nowhere explained. Above all else, a scientific paper should be understandable. 

Response 1: Thank you very much for your approval of the manuscript. I am very sorry for the confusion caused by these abbreviations in review of this manuscript. These abbreviations had been used in full names when they first appeared in the manuscript.

Point 2: Line 19 in the Abstract needs editing.

Response 2: Thank you very much for your advice. I have revised in the manuscript.

Point 3: One line of explanation of the HepG2 and BEL-7402 cell lines would be helpful, giving the tissue type and explaining the key differences.

Response 3: HepG2 (human hepatoma cell,ATCC HB-8065) and Bel7402 (human hepatoma cell, SNL-148). Although they are both derived from primary liver cancer tissues, they have some common features, such as AFP is positive and can secrete some plasma proteins. Both subcutaneous and intrahepatic inoculation can increase the tumorigenicity of nude mice. However, they are very different in drug screening, apoptosis mechanism and drug resistance.

Point 4: On line 212 the paper mentions melanoma cells, which appear nowhere else in the paper.

Response 4: Thank you very much for your careful review. In the discussion, we introduced this reference to show that PG could cause the degradation of anti apoptotic protein Bcl-2 in other cells.

Point 5: The TUNEL assay is unknown to me.

Response 5: TUNEL (TdT-mediated dUTP Nick-End Labeling) assay could detect apoptosis. When genomic DNA breaks, exposed 3 '- OH can be detected by fluorescence microscopy or flow cytometry by adding fluorescein dUTP under the catalysis of Terminal Deoxytrophic Transfer (TdT). This is the principle of TUNEL assay to detect apoptosis.

Point 6: What is 4-PBA, and why was it added? What effect was predicted, and did this happen?

Response 6: 4-PBA is an inhibitor of endoplasmic reticulum stress (ERS). Our study found that PG could cause ERS and apoptosis in liver cancer cells, but whether apoptosis is through the ERS pathway, so 4-PBA was introduced to inhibit the occurrence of ERS in liver cancer cells, and it was found that apoptosis was reduced.

Point 7: The statistics demonstrating significance are sometimes missing. For example it is stated on Page 6, line 144, that the tumor volume and weight were significantly smaller in the treatment group, but how many tumors were examined? What are the standard deviations on the inhibition rates?

Response 7: Thank you very much for your careful review. I'm sorry I was careless enough to miss the standard deviation. There were six nude mice per treatment group, and I had supplemented the standard deviation in the revised manuscript.

Point 8: Figure 4B is completely unexplained, with the legend simply repeating a line of the text. What does this figure show, and why is it important? I do not understand a single acronym on the vertical axes of these graphs.

Response 8: I am very sorry for the difficulty in reviewing the manuscript. These single acronym words were explained in the method. The levels of total aspartate aminotransferase (AST), alanine aminotransferase (ALT), alkaline phosphatase (ALP), albumin (ALB), total protein (TP), creatinine (CR), uric acid (UA), and urea in the blood represent the liver and kidney function of nude mice. The purpose of testing these indicators was to observe whether PG had an impact on liver and kidney functions, and to provide experimental basis for future clinical medication. I have illustrated that in the figure legend.

Point 9: The importance of the paper seems to be the fact that it demonstrates anti-tumor activity in vivo, while leaving healthy tissue unharmed. The major conclusion is that PG exerts its effect through stress of the endoplasmic reticulum. PG increases ER stress, and apparently therefore upregulates the expression of Bip, an ER chaperone. 4-PBA seems to reduce ER stress, and also the expression of ER chaperones. It is not described what efforts have been made to determine the cellular target(s) of PG, and whether it works in direct opposition to 4-PBA.Reading the very extensive literature on PG, I came across a paper (https://doi.org/10.1158/0008-5472.CAN-07-1919) by Konopleva and colleagues from 2008 which answered several of the questions I found myself asking, which provides a simple mechanism for the cellular effects of PG derivatives. I think the authors of the present manuscript could have saved me some time by citing it.

Response 9: Thank you very much for your advice. I had added this reference to discussion in the revised manuscript.

Point 10: On line 211, the authors state "Cytochrome C released in the mito chondrial apoptosis pathway can activate caspase9 to shear apoptosis pathway caspase3, thus inducing apoptosis [20]." Looking at the paper cited (Hosseini et al, 2013), we find in the Abstract nothing about cytochr ome c or caspases. Quoting Hosseini, we find "our results indicate a large affinity of prodigiosin for MCL-1, an anti-apoptotic member of the BCL-2 family." Wonderful! Lights switch on in my head! We have a small molecule binding to a known protein of known function, and my feet have found solid ground. Citing this paper early in the Introduction, at the end of a sentence which accurately reflects its contents, would make this paper a far, far easier read.

Response 10: Thank you very much for your advice. I have rewritten this section and quoted the references correctly.

Point 11: Clearly the PG family of molecules has been undergoing clinical trials for many years, and some mention could be made of the current state of affairs. Why have trials stopped, and what can the authors do to bring new derivatives to the clinic? Above all, why does PG exert an effect on cancer cells, and not healthy ones?

Response 11: Thank you very much for your interest in PG. Research on prodigiosin has not stopped, but the current research mainly focuses on increasing its production. If we want to introduce new derivatives into the clinic, we need to have more in-depth research on the anti-tumor mechanism of PG, which requires more experimental exploration and further research on the structure-activity relationship of PG. Why does PG exert an effect on cancer cells, and not healthy ones? It is difficult to accurately explain why it has no effect on healthy cells, but our research could partly explain it. Endoplasmic reticulum stress (ERS) refers to nutrient deficiency, pH change, hypoxia, or oxidative stress, which disrupts the folding function of the endoplasmic reticulum, causing excessive accumulation of poorly folded proteins. The poorly folded protein enters the endoplasmic reticulum, initiates unfolded protein response (UPR), and participates in several tumor biological processes, including metastasis, proliferation, and drug resistance of tumor cells. However, the ER in normal cells is in a steady state. This study initially investigated that PG could induce apoptosis in HCC cells via ERS. Therefore, PG had no effect on healthy cells.

Point 12: Overall the papers cited are a very small selection from a large field, and one wonders why recent relevant papers from the same journal (such as doi: 10.3390/molecules27123729) are not mentioned. Expanding the introduction would give the authors ample scope to describe the recent progress in this area. For the record, I have never published anything related to PG.

Response 12: It is indeed my negligence not to find this article, which provides ideas for my further research on PG derivatives. Thank you for your constructive suggestions. I have cited this article in the discussion.  

Reviewer 2 Report

The topic is very interesting and promising in the field of liver cancer research. The key findings and conclusions seem smooth, however, design and presentation must be improved.

Major issues:

-HCC is NOT the second deadliest type of cancer (see reference in the commented manuscript). REF no.1 does not support the sentence, while REF no.2 states it in the abstract, while I did not find any reference for that statement.

-figures should be reordered and revised. Axis labels missing, legends invisibly small, control expression values not at value 1.0 etc. Microscopic images are not informative in this size. Scalebars are not readable nor mentioned in the text. Strongly suggest to separate some figures with 10+ panels to two figures. Some images should be put in decent sizes to the supplement. Some figures simply cannot be evaluated in their present form.

-methods should be cleared (calculation from histogram images to bar-graphs, evaluation of colocalization, proper use of t-test or ANOVA, methods of concentration measurements from blood, etc.)

Besides, some minor suggestions have been made through the manuscript.

Author Response

Point 1: The topic is very interesting and promising in the field of liver cancer research. The key findings and conclusions seem smooth, however, design and presentation must be improved.

Response 1: Thank you very much for your careful review of this manuscript.

Point 2: HCC is NOT the second deadliest type of cancer (see reference in the commented manuscript). REF no.1 does not support the sentence, while REF no.2 states it in the abstract, while I did not find any reference for that statement. 

Response 2: Your comments are greatly appreciated. The references cited are indeed inaccurate, and I have made revisions in the revised manuscript.

Point 3: figures should be reordered and revised. Axis labels missing, legends invisibly small, control expression values not at value 1.0 etc. Microscopic images are not informative in this size. Scalebars are not readable nor mentioned in the text. Strongly suggest to separate some figures with 10+ panels to two figures. Some images should be put in decent sizes to the supplement. Some figures simply cannot be evaluated in their present form. methods should be cleared (calculation from histogram images to bar-graphs, evaluation of colocalization, proper use of t-test or ANOVA, methods of concentration measurements from blood, etc.) Besides, some minor suggestions have been made through the manuscript.

Response 3: We are grateful for these valuable comments and feel that after incorporating the your advice, the revised manuscript has been significantly strengthened and that the significance of our findings is now more evident. We have revised the manuscript, according to the comments and suggestions, and responded, point by point.

Point 4: L11: according to the SEER database, liver cancers are only sixth in the line.https://seer.cancer.gov/statfacts/html/common.html

Response 4: Thank you very much for your careful review. I have revised in the manuscript.

Point 5: L19: the --> and(Clerical error)

Response 5: I have revised this part in the revised manuscript.

Point 6: L21: detect(Clerical error)

Response 6: I have revised this part in the revised manuscript.

Point 7: L33: no it is sixth. even ref1 is not stating this.

Response 7: I have revised this part in the revised manuscript.

Point 8: Figure 1 A: ratio(Clerical error)ï¼›concentration of Bel 7402? 

Response 8: I have revised this figure in the revised manuscript.

Point 9: Figure 1 D: invisibly small

Response 9: I have revised this figure in the revised manuscript.

Point 10: Figure 1E:mark the population on the histogram which was counted as apoptotic

Response 10: The combined percentage of cells in Q2 (early apoptosis) and Q3 (late apoptosis) represents apoptotic cells, and Q1 represents dead cells. I have revised this part in the revised manuscript.

Point 11: L95:assay (Clerical error)

Response 11: Answer: I have revised this part in the revised manuscript.

Point 12: L119:this should be detailed in the methods how colocalization was found;  L134:indicate what calnexin staining is forï¼›Figure 3A:how is colocalization measured? L327:so how was the localization established.

Response 12: Calnexin is a chaperone protein that binds to the endoplasmic reticulum membrane and is an unfolded reactive protein associated with endoplasmic reticulum stress. Calnexin is activated in cells in response to biochemical, pathological, and physiological stimuli. The intracellular calnexin protein was stained with green fluorescence. Since the PG itself had a spontaneous red fluorescence, the coincidence of the two can be used to determine co-localization. I had described this in detail in the Methods section in the revised manuscript. Meanwhile, in order to visualize the results, Pearson's R value was calculated by using image J software (v1.8.0), and the results as shown in the figure 4A. Pearson's R value (Pearson's correlation coefficient) is also known as R value. The correlation coefficients range from −1 to 1. The greater the absolute value of the correlation coefficient, the stronger the correlation. The closer the correlation coefficient is to 1 or -1, the stronger the correlation is. The closer the correlation coefficient is to 0, the weaker the correlation is. Pearson's correlation coefficient is calculated as follows:

Point 13: L120:representing Ca levels?

Response 13: Yes. Fluo-3 AM is a fluorescent dye that can penetrate the cell membrane. Fluo-3 AM can be cleaved by intracellular esterase to form Fluo-3 after entering the cell, and then it can be trapped in the cell. The fluorescence intensity of Fluo-3 is sensitive to the change of Ca2+ concentration, so the fluorescence intensity can reflect the concentra tion of Ca2+ indirectly. 

Point 14: Figure 3B:how is this derived from the histograms; L135: name the axes accordingly.

Response 14: The vertical coordinates of the graph represent the intensity of intracellular fluorescence. We used fluorescence intensity to represent the calcium concentration of each group. Since the numerical representation was exponential and varies greatly, The statistical chart was made by changing the ratio of fluorescence intensity.

Point 15: Figure 3C:unreadable

Response 15: I have revised this figure in the revised manuscript.

Point 16: Figure 3D: why is control expression 0.4? if needed, clear significance signature should be shown; ## is not mentioned in the legend; L140: difference from stars?

Response 16: We used image J software to measure the fluorescence intensity of the protein bands. The ratio of target protein value to internal parameter value was used for data analysis; There were omissions in the description here. PG (5 mg/kg) vs PG (5 mg/kg) +4-PBA,#P<0.05,##P<0.01. I have revised this part in the revised manuscript.

Point 17: Figure 3E:why control bars differ from 1.0?

Response 17: I'm really sorry that the data of CHOP mRNA expression level in Figure 3E were not normalized due to improper processing in the early stage. Now we had renormalized data to get the correct statistical result graph. I have revised this part in the revised manuscript.

Point 18: L137: on (Clerical error)

Response 18: I have revised this part in the revised manuscript.

Point 19: L146:too many dots  (Clerical error)

Response 19: I have revised this part in the revised manuscript.

Point 20: Figure 4A:1.Tumor volumeï¼›2.days after first treatment

Response 20: I have revised this figure in the revised manuscript.

Point 21: Figure 4B: mention the endpoints of this measurements, which characteristics were measured; significance?

Response 21: Detection of these indicators reflects some liver and kidney function. We used these indicators to observe liver and kidney function in nude mice. Thank you very much for reminding. This statistic was omitted due to a previous oversight. Urea had statistical significance by T Test. I have revised this figure in the revised manuscript.

Point 22: Figure 4C:scale bars are unreadable, mention in fig legend. images are too small to evaluate.

Response 22: Thank you very much for your comments. Because there is no place where supplementary materials can be uploaded in the system, I have divided figure 4 into two figures.

Point 23: Figure 5A:1.not visible and not evaluatable in this form. Suggest to use lower DAPI intensity.

Response 23: Thank you very much for your constructive suggestions. We will pay attention to this problem in future experiments and reduce the intensity of DAPI. Hancong Liu who was responsible for data processing had already calculated TUNEL assay's data, but he had forgotten to put it in the composite figures in the manuscript. I have added this part in the revised manuscript.

Point 24: are the tissues from the in vivo model? if so, specify cell line and in vivo origin.

Response 24: Yes, these tumor tissues were derived from transplanted tumors in the nude mice that we treated. HepG2 (human hepatoma cell,ATCC HB-8065) cells (200 μL; 2×107 /mL) were inoculated in the right lower axilla of 36 nude mice.

Point 25: L187:phase (Clerical error)

Response 25: I have revised this part in the revised manuscript.

Point 26: L195:this belongs to results, not discussion.

Response 26: I have revised this part in the revised manuscript.

Point 27: L199:what is the suggestion with this sentence?

Response 27: HepG2 (human hepatoma cell,ATCC HB-8065) and Bel7402 (human hepatoma cell, SNL-148) come from different sources, so their genetic backgrounds are different. HepG2 cells were derived from liver cancer tissue of a 15-year-old white teenager, whereas Bel7402 was established in 1974 from clinical liver cancer surgical specimens. The chromosome number of Bel7402 cells was aneuploid with an abnormal proximal centromere chromosome. Although they are both derived from primary liver cancer tissues, they have some common features, such as AFP is positive and can secrete some plasma proteins. Both subcutaneous and intrahepatic inoculation can increase the tumorigenicity of nude mice. However, they are very different in drug screening, apoptosis mechanism and drug resistance.

Point 28: L208:expression change. (Clerical error)

Response 28: I have revised this part in the revised manuscript.

Point 29: L208:caspase levels per se are not clear markers of apoptosis, their cleavage is the key moment.

Response 29: Thank you very much for your insightful suggestion. I fully agree with you. As you said, caspase levels per se are not clear markers of apoptosis, their cleavage is the key moment. We previously focused on the apoptosis of hepatoma cells induced by PG through ERS. Our results also illustrate this point. However, the mechanism of apoptosis induced by PG through ERS is a valuable question, and we will continue to conduct in-depth research in the future.

Point 30: L249:hepatoma, HCC and liver cancer are mixed through the manuscript. Please clean the variation.

Response 30: I am very sorry for the difficulty in reviewing the manuscript. I have made changes in the revised manuscript.

Point 31: L266:from (Clerical error)

Response 31: I have revised this part in the revised manuscript.

Point 32: L270:proper company names and residency should be written at the first occurrence.

Response 32: I have revised this part in the revised manuscript.

Point 33: L274:upper index (Clerical error)

Response 33: I have revised this part in the revised manuscript.

Point 34: L275:lower index (Clerical error)

Response 34: I have revised this part in the revised manuscript.

Point 35: L277:does this mean confluency? (Clerical error)

Response 35: Yes, that means confluence. Cell fusion were mistake words I used to write this manuscript. I have revised this part in the revised manuscript.

Point 36: L292:index (Clerical error)

Response 36: I have revised this part in the revised manuscript.

Point 37: L311:cell nuclei (Clerical error)

Response 37: I have revised this part in the revised manuscript.

Point 38: L318:was (Clerical error) 

Response 38: I have revised this part in the revised manuscript.

Point 39: L330:in the dark (Clerical error)

Response 39: I have revised this part in the revised manuscript.

Point 40: L363:how?  

Response 40: I am very sorry for the trouble in reviewing paper because of the simple writing. Blood was collected from the eyeballs of nude mice, and the serum was taken after centrifugation and sent to Jiangsu Enzymatic immunity Industry Co., Ltd for testing. AST (blood volume: 15 μl), ALT (blood volume: 15 μl), ALP (blood volume: 15 μl), ALB (blood volume: 5 μl), TP (blood volume: 6 μl), CR (blood volume: 5 μl), UA (blood volume: 5 μl), and urea (blood volume: 3 μl) content in the serum was then measured by automatic biochemical analyzer (BIOBASE; BK-280 ). I have revised this part in the revised manuscript.

Point 41: L372:as for multiple group comparison, ANOVA should be used instead with an appropriate post hoc test.

Response 41: Thank you very much for your careful review. The description of statistical analysis was mistake I made in writing the manuscript. I consulted Hancong Liu  who was responsible for data processing. He said that all data were analyzed using SPSS 22.0 statistical software. Independent-sample t-test for comparison between two groups, and One-way ANOVA for comparison between multiple groups. Graphprism8 software was used for the graph. I have revised this part in the revised manuscript.

Point 42: L375:of (Clerical error)

Response 42: Answer: I have revised this part in the revised manuscript.

Reviewer 3 Report

Dear,

I would like to say that this article is very interesting in vitro study about mechanism of antiproliferative effect of the prodigiosin from Serratia marcescens.

If the above suggestions are considered, I am supportive of this work being published in Molecules.

Sincerely,

Author Response

Point 1: I would like to say that this article is very interesting in vitro study about mechanism of antiproliferative effect of the prodigiosin from Serratia marcescens. If the above suggestions are considered, I am supportive of this work being published in Molecules. I would like to say that this article is very interesting in vitro study about mechanism of antiproliferativeeffect of the prodigiosin from Serratia marcescens. There are a few technical mistakes. You should check the whole article once again and correct them.Here are some of them that I have found.

Response 1: We are grateful for these valuable comments and feel that after incorporating the your advice, the revised manuscript has been significantly strengthened and that the significance of our findings is now more evident. We responded to and revised the manuscript point by point based on these comments and suggestions.

Point 2: You do not use superscript and subscript.

Response 2: Thank you very much for your careful review. I have revised in the manuscript.

Point 3: For example it should be written Ca2+, you wrote Ca2+. Also it should be written CO2, you wrote CO2.Also it should be written 104/mL, you wrote 104/mL, it should be written mm3, and you wrote mm3.

Response 3: I have revised in the revised manuscript.

Point 4: In whole article there are a lot of technical mistakes like there is an excess of space or there is no spacebetween two words, example: line 12, 19, 64, 111, 145, 146, 160, 164,181,188, 225 etc.

Response 4: I have revised in the revised manuscript. 

Point 5: I think it should be presented structure of prodigiosin in this article. Line 69-75. It should be reference for your previous research.

Response 5: Your suggestion is very meaningful. However, since the structure of the PG isolated by our research group had been identified in another article (reference 10), it was not shown in this paper. Only a brief introduction was given in the introduction.  I am very sorry that“Our previous research” had caused misunderstanding in your review. This referred to our previous pre-experiment that discovered this phenomenon. This manuscript was a further study of this phenomenon.

Point 6: Text Figure 1. There should be point (“.”) after every capital letter, you wrote just after A.Line 99. It should be written Bcl-2, you wrote bcl-2.

Response 6: I have revised in the revised manuscript.

Point 7: Line 106: It should be written effect of PG, you wrote effect PG.

Response 7: I have revised in the revised manuscript.

Point 8: Text Figure 2. There should be point (“.”) after every capital letter, you wrote just after A. and B.

Response 8: I have revised in the revised manuscript.

Point 9: Line 140. It should be written mL, you wrote Ml. There is no word of. You wrote “3.2μg/Ml pg +” itshould be written “3.2 μg/mL of PG +”.

Response 9: I have revised in the revised manuscript.

Point 10: Line 362. Urea should be written urea.

Response 10: I have revised in the revised manuscript.

Point 11: All references should be written at the same manner. If you use abbreviation it should be written withpoints between the words

Response 11: I have revised in the revised manuscript.

Point 12: Line 402, ref 7. What is Cancer Di scovery?

Response 12: Thank you very much for your careful review. I made a mistake in the name of the magazine. I have revised in the revised manuscript.

Point 13: Ref 8 wrote whole number of pages not like this 783-94.

Response 13: I have revised in the revised manuscript.

Point 14: Ref 10. there is no space between apoptosis and inducing

Response 14: I have revised in the revised manuscript.

Point 15: line 409.Ref 12 there is no last page of used article, just first one 10-?

Response 15: Thank you very much for your careful reading. This reference has 16 pages, but the first page starts from 1. I have revised in the revised manuscript.

Point 16: Ref 14 there is no last page of used article, just first one 375-?

Response 16: This reference has 17 pages, but the first page starts from 1. I have revised in the revised manuscript. 

Round 2

Reviewer 2 Report

The revised manuscript is improved and addressed most of the reviewer's concerns. However, especially for figure refinement and axis labeling should be revised. Some methodologic clarification (as properly responded in the response to the reviewers) for TUNEL assay and 3-Fluo-AM assay are encouraged to be included in the methods part for better understanding.

After the mentioned changes in the comments, the manuscript seems suitable for publication.

Author Response

Reviewer 2

Point 1: The revised manuscript is improved and addressed most of the reviewer's concerns. However, especially for figure refinement and axis labeling should be revised. Some methodologic clarification (as properly responded in the response to the reviewers) for TUNEL assay and 3-Fluo-AM assay are encouraged to be included in the methods part for better understanding. After the mentioned changes in the comments, the manuscript seems suitable for publication.

Response 1: Thank you very much for your approval of the manuscript. Through your constructive comments, methods sections, figure refinement and axis labeling have been further revised.

Point2: L62: with strong antitumor effect

Response 2: I have revised this figure in the revised manuscript.

Point 3: L81: HCC cells

Response 3: I have revised this figure in the revised manuscript.

Point 4: Fig1A: definitely not HepG2! it is concentration of the compound!!! Like: PG concentration (ug/ml); cell line name would be more clear above the panel.

Response 4: I have revised this figure in the revised manuscript.

Point 5: Fig1C:1. rate refers to speed. This seems more like relative colony number. However, since control bars are not reaching 1.0, it is not clear for the reader. methods did not provide further information. Text size could be further enlarged. significance markers are pixelated, please reach out to editorial office about the issue to make good quality final images.2.enough once only with bigger size2.enough once only with bigger size

Response 5: In the previous calculation, the percentage of colony-forming rate was calculated to evaluate colony formation efficiency. Colony-formation rate was calculated as: (Colony counts experiment group/ Colony counts control group) × 100%. Thank you very much for your advice. As you said, rate refers to speed. I have revised it to: Number of clusters (Fold change to control) was calculated as: (Colony counts experiment group/ Colony counts medium control group) × 100% in the method section. I have also revised this figure in the revised manuscript.

Point 6: Fig. 2A: 1. invisible; 2.this is out on the margin. I suggest to follow the color pattern of Fig1. This means that Hepg2 panels would use different colors than Bel7402 figures, indicating the different treatment conditions; 3. check significance.

Response 6: I have revised this figure in the revised manuscript. 

Point 7: Fig. 2B: 1. PG; 2. channel names are not informative. rename to markers (such as annexin-V and PI). About your response no. 10: Q4 are cells alive, Q3 are early apoptotic cells, Q1 are early NECROTIC cells (no annexin positivity), and Q2 are dead cells (unknown whether via apoptosis or necrosis). (Point 10: Figure 1E:mark the population on the histogram which was counted as apoptotic.

Response 7: Thank you very much for your constructive suggestions. As you said, Q2 are dead cells (unknown whether via apoptosis or necrosis). Instead of defining Q2 as apoptotic cells, I defined Q2 and Q3 as Annexin V-positive cells. The results indicated that the toxicity of PG to HCC increased with the increase of concentration.  I have revised this part in the manuscript.

Point 8: L110: Figure legend: there are no triple stars on this panel.

Response 8: Thank you very much for your careful review. I have revised in the manuscript.

Point 9: L111: and what about **?

Response 9: I have revised this part in the revised manuscript.

Point 10: L115: ERS-inhibitor

Response 10: I have revised this part in the revised manuscript.

Point 11: Fig. 3A: 1. TUNEL is not a measure. Precise description in text and axis label is needed to se whether ration of apoptotic cells, positive pixels, or else is shown here; 2.somewhat confusing the use of the same color code that was different in former figures. Please consider uniform coloring among figures.

Response 11:  I have revised this part in the revised manuscript.

Point 12: Fig. 3C: 1. protein expression normalized to GAPDH signal. Since the scale is relative to the exposition length for each protein, I suggest rescale them that each control bars represent 1.0, and treatments are compared to that; 2. Is HepG2 Bcl-2 level not significantly increased for the black bar?

Response 12: Thank you very much for your careful review. We have defined each control bar to represent 1.0, and treatments are compared to that.

Point 13: L123. apoptosis is not really a mechanism. "Induction of apoptosis by PG treatment of HCC cells"

Response 13: I have revised this part in the revised manuscript.

Point 14: Fig. 4B: 1. Author response: The vertical coordinates of the graph represent the intensity of intracellular fluorescence. We used fluorescence intensity to represent the calcium concentration of each group. Since the numerical representation was exponential and varies greatly, The statistical chart was made by changing the ratio of fluorescence intensity. In reality, vertical axis represents event count. Fluo-3 fluorescence is on the horizontal. The sentence of deriving the bar graphs is not clear. It must be clearly stated what we see: mean vs. median fluorescence, or ratio of two fluorescence means/medians? colors on left and right panels should be parallelï¼›2. Fluo-3 intensityï¼›3. what is 100%? must be clear what the dimension is!

Response 14: I am really very sorry for the mistake in the statistical chart. I consulted with Hancong Liu, and he said that what we see was the mean fluorescence value. “%” description was wrong. He wanted to shrink the vertical axis by “102 instead of “%” in the statistical chart. I have revised this figure in the revised manuscript. We have used different colors to represent different concentrations.

Point 15: Fig. 5: "of" not needed

Response 15: I have revised this part in the revised manuscript.

Point 16: L163: HCC in the previous line.

Response 16: I have revised this part in the revised manuscript.

Point 17: L170: "main" is imprecise, name them.

Response 17: I have revised this part in the revised manuscript.

Point 18:Figure 6: HepG2

Response 18: I have revised this part in the revised manuscript.

Point 19: L186: 1. mechanism should be replaced: apoptosis inducing effect of PG; 2. HepG2

Response 19: I have revised this part in the revised manuscript.

Point 20: Figure 8A: please rename axis

Response 20:  I have revised this part in the revised manuscript.

Point 21: L197: 1. this is not the figure title. fore example the entire figure A is about TUNEL assay, unrelated to protein expression; 2. in the tumor xenografts of HepG2 HCC model.

Response 21: Thank you very much for your careful review. I have revised this part in the revised manuscript.

Point 22: L200: s

Response 22: I have revised this part in the revised manuscript.

Point 23: L281: still suggest to think about uniform mentioning, the jumps between hepatoma and HCC is confusing.

Response 23: I am very sorry for the jumps between hepatoma and HCC. I have changed hepatoma to HCC in the revised manuscript.

Point 24: L305: America is not a country.

Response 24: I have revised this part in the revised manuscript.

Point 25: L340: ERS-inhibitor

Response 25: I have revised this part in the revised manuscript.

Point 26:L346: Detailed reaction or reference for TUNEL assay (like in reviewer response) shoud be included.

Response 26: Thank you very much for your advice and I have added the appropriate references in the methods section.

[28]Guo, W.; Wang, Y.; Wang, Z.; Wang, Y.P.; Zheng, H. Inhibiting autophagy increases epirubicin's cytotoxicity in breast cancer cells. Cancer science. 2016, 11,1610-1621.

Point 27: L368: measurement of

Response 27: I have revised this part in the revised manuscript.

Point 28: L369: flou -> fluo

Response 28: I have revised this part in the revised manuscript.

Point 29: L372: Fluo-3 AM can be cleaved by intracellular esterase to form Fluo-3 after entering the cell, and then it can be trapped in the cell. The fluorescence intensity of Fluo-3 is sensitive to the change of Ca2+ concentration, so the fluorescence intensity can reflect the concentra tion of Ca2+ indirectly. Detailed explanation may be excluded if a proper reference is added.

Response 29: Thank you very much for your constructive suggestions. I added a proper reference in the method section to illustrate that Fluo-3 fluorescence intensity is sensitive to Ca2+ concentration changes in the revised manuscript.

[29] Wang, J.; Ming, H.; Chen, R.; Ju, J.M.; Peng, W.D.; Zhang, G.X.; Liu, C.F. CIH-induced neurocognitive impairments are associated with hippocampal Ca(2+) overload, apoptosis, and dephosphorylation of ERK1/2 and CREB that are mediated by overactivation of NMDARs. Brain. Res. 2015, 1625, 64-72.

Point 30: L395: is this an abbreviation for some platinum compound? why was this group started? please give details.

Response 30: I am very sorry for the trouble in reviewing paper because of the simple writing. DDP is cis-platinum (DDP). DDP can inhibit the growth of many tumor cells in clinic. In this manuscript, we used DDP as a positive group to observe the inhibition of DDP on transplanted tumor (HepG2 cells) in nude mice, which was only used as a reference to inhibit tumor growth. I have revised this part in the revised manuscript.

Point 31: L414: GraphPad Prism 8 (GraphPad Software Inc.)

Response 31: I have revised this part in the revised manuscript.

Point 32: L415: creation

Response 32: I have revised this part in the revised manuscript.
